# Genome-Wide Association of Stem Carbohydrate Accumulation and Remobilization during Grain Growth in Bread Wheat (*Triticum aestivum* L.) in Mediterranean Environments

**DOI:** 10.3390/plants10030539

**Published:** 2021-03-12

**Authors:** Fernando P. Guerra, Alejandra Yáñez, Iván Matus, Alejandro del Pozo

**Affiliations:** 1Instituto de Ciencias Biológicas, Universidad de Talca, Talca 3460000, Chile; fguerra@utalca.cl; 2Centro de Mejoramiento Genético y Fenómica Vegetal, Facultad de Ciencias Agrarias, Universidad de Talca, Talca 3460000, Chile; myanez@ucm.cl; 3Facultad de Ciencias Agrarias y Forestales, Universidad Católica del Maule, Talca 3460000, Chile; 4Centro Regional de Investigación Quilamapu, Instituto de Investigaciones Agropecuarias, Chillán 3780000, Chile; imatus@inia.cl

**Keywords:** carbohydrate distribution, water deficit, grain filling, wheat breeding, genome, Mediterranean climate

## Abstract

Water deficit represents an important challenge for wheat production in many regions of the world. Accumulation and remobilization of water-soluble carbohydrates (WSCs) in stems are part of the physiological responses regulated by plants to cope with water stress and, in turn, determine grain yield (GY). The genetic mechanisms underlying the variation in WSC are only partially understood. In this study, we aimed to identify Single Nucleotide Polymorphism (SNP) markers that account for variation in a suite of WSC and GY, evaluated in 225 cultivars and advanced lines of spring wheat. These genotypes were established in two sites in the Mediterranean region of Central Chile, under water-limited and full irrigation conditions, and assessed in two growing seasons, namely anthesis and maturity growth periods. A genome-wide association study (GWAS) was performed by using 3243 SNP markers. Genetic variance accounted for 5 to 52% of phenotypic variation of the assessed traits. A rapid linkage disequilibrium decay was observed across chromosomes (*r*^2^ ≤ 0.2 at 2.52 kbp). Marker-trait association tests identified 96 SNPs related to stem weight (SW), WSCs, and GY, among other traits, at the different sites, growing seasons, and growth periods. The percentage of SNPs that were part of the gene-coding regions was 34%. Most of these genes are involved in the defensive response to drought and biotic stress. A complimentary analysis detected significant effects of different haplotypes on WSC and SW, in anthesis and maturity. Our results evidence both genetic and environmental influence on WSC dynamics in spring wheat. At the same time, they provide a series of markers suitable for supporting assisted selection approaches and functional characterization of genes.

## 1. Introduction

Bread wheat (*Triticum aestivum* L.) is one of the world’s most important cereal crops in terms of cultivated area and economic activity [1]. It is expected that global demand for bread wheat will increase by 70%, by 2050, in line with the projected growth in world population [2]. Drought and high temperatures are the most important abiotic factors affecting nearly 200 million ha of global wheat production [3]. Wheat is very sensitive to climatic and environmental conditions [4], which is reflected in different yields when the same genotype is grown under water-limited and full irrigation conditions [5].

In Mediterranean-climate regions, temperate cereals like wheat are usually exposed to a severe water deficit during the grain filling period, the so-called terminal drought-stress [6,7,8,9]. This water deficit leads to reductions in canopy photosynthesis, as well as lower levels of assimilate transfer to the grain, leading to smaller kernel weights and GY [7,10]. Water-soluble carbohydrates (WSCs) are accumulated in stems during pre-anthesis (and up to 20% of them after anthesis in wheat) as reserves, and later these are remobilized to the grain [11,12,13], reducing the negative effects of water deficit. Indeed, the contribution of stem WSCs to grain growth could be very significant under water-deficit conditions [14]. A wide genetic variability in stem WSCs has been reported for wheat. Considering that variability, high and low WSC genotypes with differential yield, stem biomass, or yield tolerance index have been identified, utilizing recombinant inbred lines [12], cultivars and advanced semi-dwarf lines [13,15,16], lines deriving from synthetic hexaploids [14], and double-haploids populations [17]. The variation in total WSC has been attributed to genotypic variation in components of the fructan biosynthesis [16,18].

Physiological traits associated with tolerance to water deficit, such as accumulation and remobilization of stem WSCs, are included in some breeding programs, to increase yield in rainfed environments [19,20]. These traits are quantitative, determined by multigenic control, and strongly influenced by the environment [21]. The “dissection” of complex traits has been enhanced by the development of analysis technologies focused on identifying quantitative trait loci (QTL) underlying phenotypic variation [22,23,24,25]. Analysis of candidate genes and genome-wide association studies (GWASs) have been performed in bread wheat to identify SNP markers controlling traits such as dehydration tolerance [26,27], leaf rust resistance [28], grain yield [29], carbon isotope discrimination [30], kernel and root morphology [31,32], resistance to low nitrogen [33], and leaf traits [34], among others. For instance, a GWAS analysis of stem WSCs at their maximum concentration (~14 days post-anthesis) was performed in 166 Chinese bread wheat cultivars (grown in four environments), using 18,207 SNP markers, and this identified 52 significant marker-trait associations on all wheat chromosomes except for 2A, 2D, 4D, 5B, 6A, and 6D [35]. Another study conducted in a large collection (*n* = 990) of bread wheat genotypes in Australia revealed significant associations between stem WSCs at post-anthesis and molecular markers (SNPs and diversity arrays technology (DArT)) located on chromosomes 1A, 1B, 1D, 2D, and 4A [36]. Recently, Fu et al. [37] conducted a GWAS on 166 wheat accessions, from China and other countries, utilizing SNP arrays. They identified 19–36 significant loci associated with WSC (measured by near-infrared spectroscopy) at 10 to 30 days post-anthesis. These loci were distributed across the seven groups of chromosomes.

In the present study, we performed a GWAS and a haplotype analysis, to identify SNP markers accounting for variation in a suite of stem traits, including weight, the WSC content at anthesis and maturity, and the apparent remobilization of stem WSCs. We analyzed 225 advanced spring wheat lines, which were established at two sites in the Mediterranean-climate region of Central Chile, under water-limited and full irrigation conditions, respectively. These wheat genotypes were obtained from different breeding programs developed by research institutions in Chile, Uruguay, and Mexico. We hypothesized that, in a diverse set of accessions, a high-density genotyping with SNP markers and the linkage disequilibrium present in specific regions of the genome would allow identification of new genome-phenotype associations for stem carbohydrate traits related to tolerance to water deficit. This study complements our previous research [15,16,30] on the genetic and genomic characterization of physiological and agronomical traits of bread wheat under water deficit. At the same, considering the importance of the genotype by environment interaction on wheat yield and adaption, this study aimed to broaden our current knowledge about the marker-trait associations, particularly under specific conditions representative of the South American Mediterranean climate. Our study analyzed that sort of associations, integrating two analytical approaches, single-marker and haplotype association tests. This topic has been scarcely described for other studies dealing with WSC and yield traits in wheat. The SNP-markers identified by these approaches contribute to defining candidate genes suitable for improving, by traditional or biotechnological breeding methods, both adaptive and agronomic traits in wheat. This will be required to keep its productivity under more challenging environmental conditions.

## 2. Results

### 2.1. Grain Yield, Stem Carbohydrate Accumulation, and Remobilization Were Influenced by Genotype and Site

We compared the 225 wheat cultivars and advanced lines grown at two sites during 2012, to analyze the effect of contrasting water regimes on genotype performance. These sites, Cauquenes and Santa Rosa, represented water-limited (rainfed) and full irrigated conditions, respectively. Large differences in GY were observed between both sites, with an average of 3140 ± 550 and 9820 ± 970 kg ha^−1^ at Cauquenes and Santa Rosa, respectively (Figure 1a). The opposite pattern was observed for SWa, WSCa, and ARWSC (Figure 1b–f), indicating an increase in these traits in response to a lower water availability (data distributions for the other traits are in Appendix A). Mean values for the different traits in each site conditions are given in Appendix A. Analysis of variance revealed a significant genotype by site effect for GY, SWm, WSCm, WSCCm and WSCCa/m^2^ (Table 1). For the other traits, the genotype (G) and site (S) main factors were both significant. The relative importance of the genotype by site (GxS) interaction was reflected in the percentage of phenotypic variance that this factor accounted for, ranging from 12.4 to 18.3% for SWm and WSCCa/m^2^, respectively. The genetic variance, estimated by *σ*^2^*_G_*, ranged from 5.0 to 51.6% of the phenotypic variation for WSCm and SWa, respectively.

### 2.2. Significant SNPs Were Detected in Association with Stem Carbohydrate Accumulation and Remobilization

The single-SNP tests identified ninety-six significant associations across all sites (Table 2). The total number of significant SNPs varied according to the trait analyzed, ranging from 2 (for SWa and SWm) to 23 (for WSCa), but it also varied depending on the site, from 5 (Cauquenes 2011) to 36 (Santa Rosa FI 2011). The identified SNPs were specific for each site and year, as is depicted in Appendix A. Significant SNPs were distributed across almost all the wheat genome, with a higher representation of 6B and 6D chromosomes, which had 10 and 14 SNPs, respectively. Details of the significant SNPs per chromosome are provided in Appendix A.

The highest proportion (64%) of significantly associated SNPs was located within intergenic regions, whereas nearly a third of them (34%) were part of gene coding regions (Figure 2a); for this last category, 37% encoded proteins without a characterized function and 63% were proteins participating in energy/metabolism (18%), protein synthesis/modification (12%), transcription (12%), and signal transduction, among others (Figure 2b).

A list of SNP-markers significantly associated (*p*-value < 0.001 and *q*-value < 0.15) with the assessed traits is presented in Table 3 (a complete list is included in Appendix A). For WSCa in Cauquenes, in 2012, the detected SNPs belonged to genes encoding a serine/threonine-protein kinase, a guanosine nucleotide diphosphate dissociation inhibitor, a strictosidine synthase, an ABC transporter C family member 10, and a hypersensitive induced response protein 3. These proteins are part of functional categories, such as protein synthesis or modification, energy or metabolism, cell transport, and pathogen defense. Other SNPs, located in genes encoding uncharacterized proteins, were also identified for WSCa in Cauquenes, 2012, and ARWSC in Santa Rosa, 2011. Finally, for ARWSC and WSCCa/m^2^ in Santa Rosa, 2011, some significant SNPs were also located in intergenic regions. In terms of the specific effect of each SNP on trait variation (*R*^2^), it ranged from 8.4% (0.0837) to 14.9% (0.1491).

### 2.3. A Rapid Linkage Disequilibrium (LD) Decay Was Observed across Chromosomes

The extent of linkage disequilibrium (LD) was analyzed across each chromosome. On average, LD over physical distance decayed to *r*^2^ ≤ 0.2 at 2.52 kbp. The variation in the decay depended on the specific chromosomes, with the most rapid decay observed on chromosome 2D (*r*^2^ ≤ 0.2 at 1.09 kbp) and the slowest decay on chromosome 6D (*r*^2^ ≤ 0.2 at 8.31 kbp). Graphs showing the LD decay per chromosome are included in Appendix A.

The LD observed for two representative genomic regions containing significant SNPs is depicted in the heatmaps in Figure 3. The first corresponds to an intergenic region (1870 bp) located on chromosome 6D. This region contained the markers iniaGBS52443 (SNP1382) and iniaGBS52444 (SNP1383), which were significantly associated with WSCCa and WSCCa/m^2^ (Figure 3a, Table 3, and Appendix A). The second region (768 bp) is on chromosome 3A, and it encodes a serine/threonine-protein kinase. A marker within this gene, iniaGBS56801 (SNP1507) was associated with WSCa (Figure 3b). These SNPs, in an intergenic and gene region, were part of moderate-to-high LD and low LD regions, respectively.

### 2.4. Single-SNP Markers Are Part of Significant Haplotypes in Association with Traits

As an insight to determine the effect of haplotypes on the studied traits, we carried out an SNP-haplotype analysis for three gene regions harboring significant single-SNP markers. For the serine/threonine-protein kinase gene (chromosome 3A), we identified four significant haplotypes that were associated with SWa and another three associated with SWm, WSCCm, and WSCCa in Cauquenes, 2012 (Table 4). Analysis of the intergenic region on chromosome 6D identified three significant haplotypes for SWa, three for SWm, and one for WSCCm/m^2^, in Santa Rosa, 2011 (full irrigation). Finally, for the gene encoding an endoglucanase, present on chromosome 7B, we identified two significant haplotypes in association with WSCa and WSCCa/m^2^, in Santa Rosa, 2011 (mild water stress). From these results, we further considered SWa as a study case, to analyze the specific effect of haplotypes on phenotype. In Santa Rosa, 2011 (full irrigation), haplotypes H1 (ACACCAGGGCCT) and H3 (CCGTTAGGGTTC), belonging to the intergenic region, differed significantly in terms of their mean values for SWa (Figure 4a). Similarly, for Cauquenes 2012, the H1 (GTGTTCAT) and H4 (GTAGCTGT) haplotypes in the serine/threonine-protein kinase gene had a significantly lower mean SWa than H2 (GTAGCTAC) and H3 (TTAGCCAT) (Figure 4b).

## 3. Discussion

The GWAS analysis of GY and nine stem WSC traits was performed in a panel of 225 advanced wheat lines and cultivars grown in environments subjected to terminal-drought-stress and full-irrigation conditions during two growing seasons. Although the selected genotypes had a relatively narrow range of heading dates (from 79 to 84 days), a significant effect of genotype, site, or GxS interaction was observed on all the analyzed traits (Table 1). These effects confirmed the high phenotypic variability for physiological and agronomic traits estimated previously, when a larger set of genotypes (378 lines, including those included in this study) was analyzed [15]. Furthermore, in an elite wheat-breeding population assessed in Australia [38], genotype and site, either as principal or combined (GxS) factors, were also shown to be significant for WSCC and WSCC/m^2^ (at approximately 180 degree-days post-anthesis) across environments differing in water availability. In the current work, the differential influence of genetics relative to other factors (including environmental ones) was represented by a low-to-moderate *σ*^2^*_G_/σ*^2^*_P_* ratio, which ranged from 0.05 to 0.52 for WSCm and SWa, respectively (Table 1). These estimates are lower than others mentioned by Dong et al. [35] for WSC, indicating that this trait is largely genetically determined (broad-sense heritability of 0.78–0.90). However, distinct genotypes, experimental conditions, genetic analyses, and methods for measuring WSC would explain the observed difference between studies.

Genotyping and analysis of LD detected a relatively rapid decay in this parameter over physical distance (*r*^2^ ≤ 0.2 at 2.52 kbp). A similar pattern was reported by Sela et al. [39] for emmer wheat (*Triticum dicoccoides*), which decayed from *r*^2^ = 0.5 to *r*^2^ = 0.1 within a 2 kbp span. These authors suggested a significant effect of recombination operating as a diversifying force. The breakdown in LD is primarily driven by recombination, so the extent of LD is expected to vary as an inverse relation to the local recombination rate [40]. In this study, we considered a density of 0.22 SNPs per Mbp (Appendix A), after SNP filtering. Considering an average of 14.6 SNPs per Mbp estimated in a recently published wheat genome sequence [41], further studies should be able to confirm our estimation for LD decay.

The GWAS analysis using 3243 SNP-markers identified 96 significant associations for the assessed traits (Table 2). Significant single-SNPs were specific to each site, water regime, and growing season (Appendix A), showing an experiment-specific pattern of results, and the presence of an overall GxS and genotype by time (GxT) effect (between growing seasons). A similar significant involvement of GxS in phenotype variability in wheat was identified for WSC and WSCC in experiments conducted in China [35] and Australia [36], respectively. In our study, the number of significant associations depended on the trait (Table 2) and the chromosome (Appendix A). The contribution of particular chromosomes within the wheat genome towards yield (e.g., GY) and WSC traits is variable within the literature describing GWAS results from trials established on sites with water limitations [30,35,37,42,43]. Indeed, most of the wheat chromosomes were represented across these studies, and the effect of individual significant SNPs on phenotypic variation, expressed in terms of *R*^2^, was spread across a wide range of values (up to 16%).

Our association tests showed that most of the significant SNPs (64%) were within intergenic regions (Figure 2). This is consistent with the overall distribution of SNPs determined from information contained in public databases (e.g., National Center for Biotechnology Information, NCBI), where about 60% of SNPs are in genomic regions outside coding regions [44]. Such polymorphisms could be part of cis-regulatory elements and modulate gene expression at a transcriptional level. In contrast, 34% of significantly associated SNPs were located within genes which, for example, encode proteins such as serine/threonine protein kinase, strictosidine synthase, ABC transporter, and hypersensitive-induced response protein, among others. These types of proteins have been implicated in responses to stress, membrane transport, signal transduction, and cell death. Other GWAS analyses of wheat WSC traits have identified candidate genes from two main categories: carbohydrate metabolism and stress responses [35,37]. Differences among studies are expected due to the diversity of experimental factors and the genotyping approaches implemented. Previously, we studied the stem carbohydrate dynamics and expression of genes involved in fructan accumulation and remobilization during grain growth [16]. In specific, we detected significant differences (at the transcript level) between tolerant and susceptible genotypes, under water stress, in the regulation of genes encoding fructan biosynthesis enzymes, such as fructan 1-fructosyltransferase (1-FFT), fructan:fructan 6G-fructosyltransferase (6G-FFT), sucrose:fructan 6-fructosyltransferase (6-SFT), fructan 1-exohydrolase *w2* (1-FEH *w2*), and fructan 1-exohydrolase (1-FEH). These genes have been mapped in different chromosomes. The 1-FFT-A1 gene was located on chromosome 4A [45]; the 1-FEH *w1*, *w2*, and *w3* genes were mapped on chromosomes 6A, 6B, and 6D [46]; and 1-SST genes were isolated and located on chromosomes 4A, 7A, and 7D [47] of Chinese bread wheat cultivars. However, after the quality control performed on genotyping data, no SNPs from these genes were represented in the datasets utilized in our association tests. The use of genotyping 90 K arrays has recently confirmed the presence of significantly associated SNPs, belonging to those genes, validating the importance of genes regulating fructan metabolism in the response of wheat to water stress [37].

From single-SNP associations (Table 3), a set of genes was identified as candidates for further functional characterization. The first of these is a gene that encodes a serine/threonine protein kinase. This enzyme has important functions in abiotic and biotic stress-signaling pathways [48]. In wheat, this type of protein has been identified in response to drought [49]. Members of the wheat SnRK gene family (e.g., *TaSnRK2.4*) have been implicated in the regulation of enhanced osmotic potential, strengthening tolerance to drought, salt, and freezing stresses [50]. The second gene encodes a guanosine nucleotide diphosphate dissociation inhibitor (GDI), a protein interacting with Rab GTPases, which mediate the targeting and fusion of vesicles in the exocytic and endocytic pathways [51]. In plants, GDI has been related to regulation of the growth of pollen tubes and root hairs, as well as tolerance to oxygen deprivation [52]. The third gene encodes a strictosidine synthase (STR), an enzyme that plays an important role in the biosynthesis of terpenoid indole alkaloids, and in rice it is important for anther development and pollen-wall formation [53]. In *Catharanthus roseus*, STR is involved in the response to salinity, low temperature stress [54], and PEG-induced drought stress [55]. The fourth gene encodes an ABC transporter C family member 10, a protein that uses ATP to transport substances across the cell membrane. In wheat, this class of transporter has been localized in the vacuolar membrane and is involved in grain formation and mycotoxin tolerance [56]. The fifth gene encodes the hypersensitive induced response protein 3, which is part of the hypersensitive reaction in plants. It is related to the development of spontaneous lesions in leaves in defense against pathogen attack, and specifically, this protein triggers cell death in maize, barley, and rice [57]. Finally, the last of these candidate genes encodes an endoglucanase belonging to pathogenesis-related (PR) defense proteins, which are accumulated in response to viruses, bacteria, and fungi, as well as to abiotic factors. In wheat, in particular, endoglucanase is produced in response to drought [58].

We performed an SNP-haplotype analysis for three gene regions harboring significant single-SNP markers. This approach allowed for the detection of wider genomic regions associated with the different phenotypes analyzed (Figure 3 and Figure 4 and Table 4). Haplotype-based analysis has been proposed as one of the most important developments in the genetic analysis of crop genomes [59]. Among other features, haplotype-analysis could capture epistatic interactions between SNPs at a locus, reduce the number of tests, and provide more power than a single marker when an allelic series exists at a locus [60]. In our study, significant haplotypes were identified in the genome in both coding and selected intergenic regions. At the same time, the haplotypes included SNPs that were identified by single-marker associations, evidencing the consistency between both approaches. Information derived from both of the detection techniques allowed for the definition of regions with high LD that were significantly associated with WSC traits, and this indicated significant differences in the performance of genotypes containing specific haplotypes (Figure 4). A haplotype-based approach also can be an alternative for detecting genome-trait associations when detection by single-SNP analysis produces a limited number of significant associations.

## 4. Materials and Methods

### 4.1. Association Population and Growth Conditions

A collection of 225 different genotypes of spring wheat, generated by breeding programs in Uruguay (Instituto Nacional de Investigación Agropecuaria, INIA), Chile (Instituto de Investigaciones Agropecuarias, INIA), and Mexico (International Wheat and Maize Improvement Centre, CIMMYT), was utilized as an association population. This set is part of the 384 lines described and analyzed previously by del Pozo [15], in terms of their performance under water-limited conditions in a Mediterranean climate. The 225 genotypes were selected by excluding lines with extreme phenological variation, considering an average for heading date of between 79 and 84 days.

Wheat genotypes were assessed in two sites located in the Mediterranean climate region of central Chile: Cauquenes (35°58′ S, 72°17 W, 177 m.a.s.l.) and Santa Rosa (36°32′ S, 71°55′ W, 220 m.a.s.l). Cauquenes is a rainfed area and represented the water-limited conditions; annual precipitation was 580 and 600 mm in 2011 and 2012, respectively. Santa Rosa represented full irrigation (FI) and moderate water stress (MWS) conditions, which were achieved through support irrigation. Trials were assayed during two consecutive growing seasons, 2011 and 2012, except for the MWS trial, which was only set up during 2011. Cauquenes corresponds to the Mediterranean drought-prone area of Chile. According to del Pozo et al. [15], the average annual temperature is 14.7 °C, the minimum average is 4.7 °C (July), and the maximum is 27 °C (January). The average evapotranspiration is 1200 mm, and the annual precipitation was 410 and 600 mm in 2011 and 2012, respectively. No irrigation was applied in this site during 2011 and 2012. Santa Rosa corresponds to a high-yielding area. The average annual temperature in this area is 13.0 °C, the minimum average is 3.0 °C (July), and the maximum is 28.6 °C. The annual precipitation was 736 and 806 mm, in 2011 and 2012, respectively. Details about the trial establishment, experimental design, fertilization, control of weeds, irrigation, and soil water monitoring were previously detailed by del Pozo et al. [15]. Briefly, plants were grown in 2 × 1 m plots in rows separated by 0.2 m, with seeds sown at 20 g/m^2^. The experiment was established as an alpha-lattice design with 20 incomplete blocks, and 20 genotypes per block. A replicate comprised 400 plots and 384 genotypes. Two replicates were included in both trials, except for the MWS treatment in 2011, at both Cauquenes and Santa Rosa, where only single replicates were established.

### 4.2. Agronomic and Physiological Traits

Stem weights at anthesis (SWa) and maturity (SWm) were evaluated in five main stems per plot per genotype. The concentration of WSC in stems was determined at anthesis (WSCa) and maturity (WSCm), using the anthrone reactive method [61]. The WSC content (WSCC) was determined per unit of stem at anthesis (WSCCa = WSCa × SWa) and maturity (WSCCm = WSCm × SWm), and on a growth-area basis (m^2^) for the same phenological stages (WSCCa/m^2^ and WSCCm/m^2^). The apparent WSC remobilization (ARWSC) was calculated as the difference in WSC content between anthesis and maturity. Grain yield (GY) was determined by harvesting the whole plot.

### 4.3. Statistical and Genetic Analysis

An approach based on a mixed linear model was applied to assess the genetic variation underlying the different traits. Initially the normal distribution was evaluated. A transformation utilizing natural logarithms was used to normalize WSC, WSCC, and WSCC/m^2^, for both the anthesis and maturity measurements. An ANOVA was carried out for each trait, using the following simplified model:*y_ijkl_ = µ + G_i_ + S_j_ + R(S)_k(j)_ + GxS_ij_ + e_ijkl_*(1)
where *y_ijkl_* is the response measured in the *l*-th plot of the *i*-th genotype within the *j*-th site and the *k*-th replicate, *μ* is the overall mean, *G_i_* is the random effect of genotype *i*, *S_j_* is the fixed effect of site *j*, *R(S)_k(j)_* is the fixed effect of replicate *k* within site *j*, *GxS_ij_* is the random interaction effect of genotype *i* with site *k*, and *e_ijkl_* is the random error in replicate *k* at site *j* within genotype *i*. ANOVA was executed by using the GLM procedure in SAS 9.2 [62]. Additionally, variance components were estimated by Restricted Maximum Likelihood, using the MIXED procedure. An estimation of the relative importance of genetic variation (*σ*^2^*_G_*) on phenotypic variation (*σ*^2^*_p_*) was estimated by the *σ*^2^*_G_**/σ*^2^*_p_* ratio, where *σ*^2^*_p_ = σ*^2^*_G +_ σ*^2^*_GxS +_ σ*^2^*_e_*, and *σ*^2^*_GxS_* and *σ*^2^*_e_* are the variance components estimated for the *GxS* interaction and residual, respectively. These analyses were done only with data from Cauquenes and Santa Rosa, for the 2012 season, considering that both datasets included two full replicates.

### 4.4. Genotyping

Genotyping by sequencing (GBS) was performed, using the Illumina HiSeq 2000 sequencing platform and the bioinformatic approach for SNP identification described previously by Lado et al. [63]. SNP-markers with a minor allele frequency < 5% and departure from Hardy–Weinberg equilibrium, as well as those with more than 5% missing data, were removed for the subsequent analysis stages, utilizing JMP Genomics 7.1 (SAS Institute, 2013a). The depurated database included 3243 SNP-markers selected from an initial dataset comprising 8746 SNPs.

### 4.5. Linkage Disequilibrium

Linkage disequilibrium (LD) was estimated among pairwise combinations of SNPs per chromosome. It was expressed in terms of the squared correlation of allele frequencies *r*^2^. The *r*^2^ value between pairs of SNP markers, within each chromosome, was estimated, using TASSEL 5.2 [64], with the option of sliding window approach (120 SNPs per window). To assess the extent of LD, the decay of LD within physical distance (base pairs) between SNPs, within each chromosome, was evaluated by nonlinear regression analysis of *r*^2^ values [65]. Analysis was performed by applying the NLIN procedure in SAS 9.4 [62].

### 4.6. Association Analysis

SNP-marker trait association tests were performed using a mixed linear model (MLM), adjusted for population structure and genetic relatedness, utilizing using TASSEL 5.2 [64]. The population structure matrix (Q) was obtained by principal component analysis (PCA), whereas the genetic relatedness (kinship) was incorporated as a scaled identity-by-descent matrix (K). Accession means were adjusted by Best Linear Unbiased Predictor, using Proc MIXED in SAS v9.2 [62]. The significance associated with the results of the SNP-marker trait association tests were subjected to multiple testing correction by false discovery rate (FDR) implemented through the program QVALUE in R [66,67].

### 4.7. Haplotype Analysis

Association tests between traits and selected genomic regions containing significantly associated single-SNPs were carried out, using the Haplo.stats package in R [68]. Haplotypes were estimated for each selected genomic region, and their frequencies were determined by using the modified expectation maximization method of haplotype inference [69]. Haplotypes and associations with traits were determined by utilizing genotyping, trait, and PCA matrices for the genotype population. Singleton alleles were discarded, as well as haplotypes with a frequency < 0.05. A global score statistic and haplotype-specific scores were derived from generalized linear models. A correction for multiple testing was also done by FDR, similar to single marker association tests. Significant differences among least-square means of haplotype classes were determined by ANOVA and the Tukey HSD test, using the GLM procedure in SAS 9.4 [62].

### 4.8. Gene Models and Annotations

Gene models and annotations for genes containing significant SNPs were obtained from the *Triticum aestivum* genome sequence available at the Ensembl platform [70]. Functional classes were defined using Quick GO [71].

## 5. Conclusions

Wheat is a key crop for world food sustainability, so genetic improvement in yield for sites with limited water availability must be a priority under the climate-change scenarios projected for this century. In that framework, identification and characterization of the genetic mechanisms underlying adaptive and productive traits are fundamental. In this study, we identified a series of SNPs controlling the variation in WSC traits in genotypes grown under different water regimes, in the Mediterranean-climate region of Central Chile. Most of these significant SNPs are part of gene-encoding proteins involved in the defensive response to drought and biotic stress. These SNPs represent a starting point to define candidate genes for functional characterization of their biological role under water stress. Our results also showed a complex and polygenic inheritance of the assessed traits. In that sense, the relative influence of genetic or environment (sites and irrigation conditions) factors was variable, depending on traits. Results from this study provided a set of SNPs that could be utilized to support marker-assisted selection suitable for future genetic improvement of spring wheat lines under water-limited areas.

## Figures and Tables

**Figure 1 plants-10-00539-f001:**
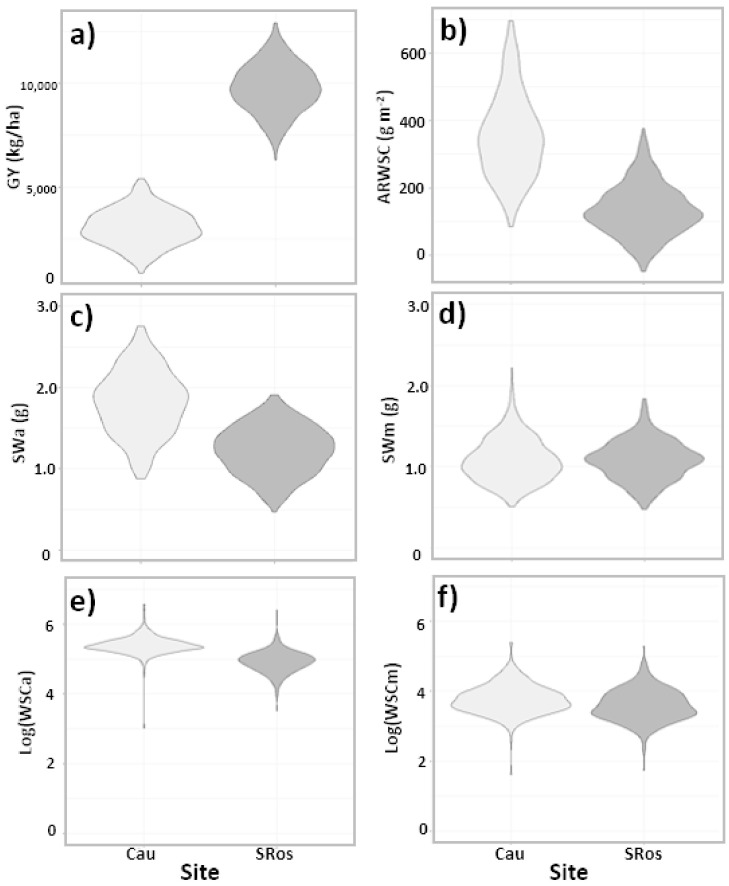
Data distribution for a selected set of traits, measured at Cauquenes (Cau; water-limited) and Santa Rosa (SRos; full irrigation) in 2012. Violin plots represent the data distribution of 225 genotypes. GY, grain yield (**a**); ARWSCs, apparent remobilization of water-soluble carbohydrates (**b**); SWa, stem weight at anthesis (**c**); SWm, stem weight at maturity (**d**); WSCa, water-soluble carbohydrates at anthesis (**e**); WSCm, water-soluble carbohydrates at maturity (**f**).

**Figure 2 plants-10-00539-f002:**
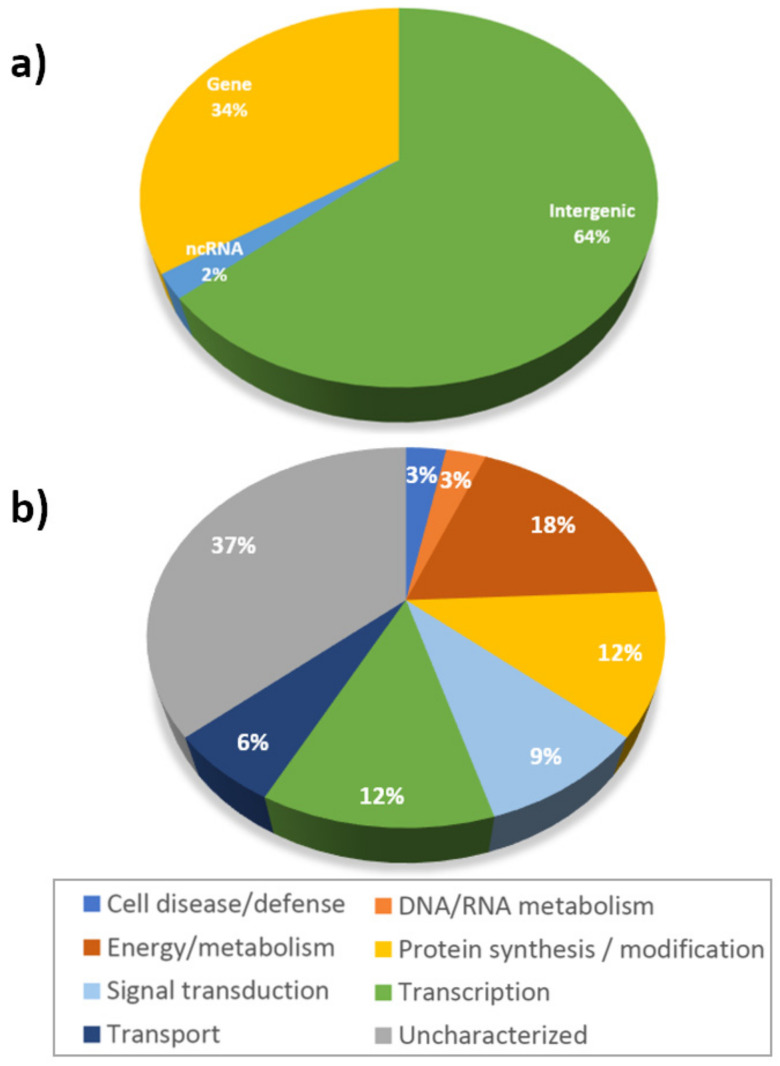
Classification of significant SNPs according to their genome location (**a**) and functional category (**b**).

**Figure 3 plants-10-00539-f003:**
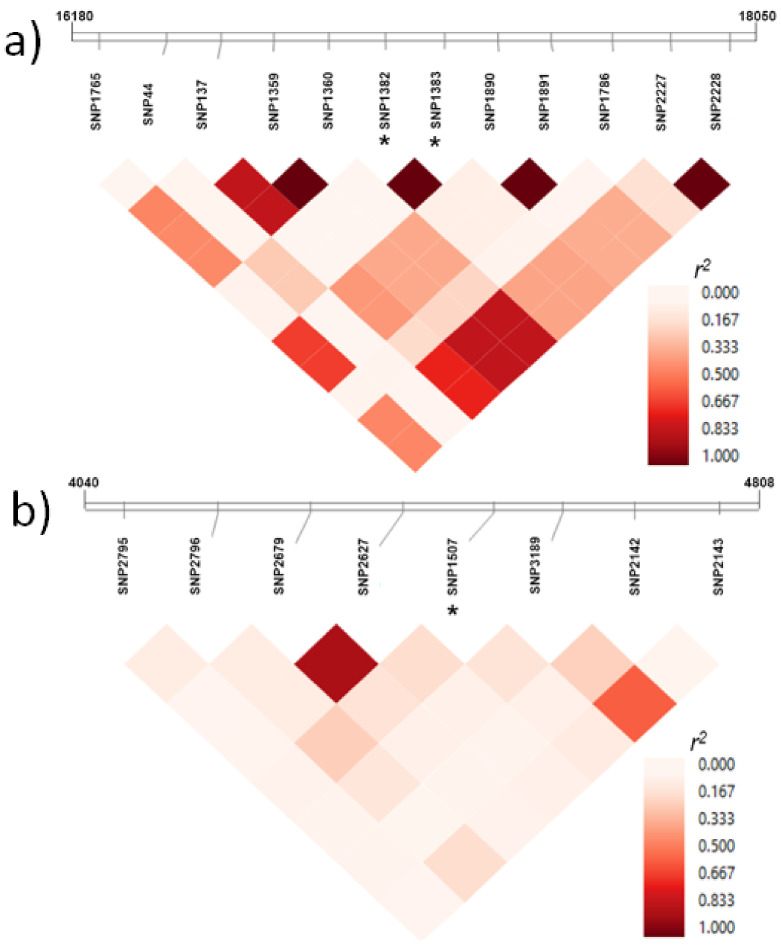
Linkage disequilibrium heatmaps for two representative genomic regions containing significantly associated single-SNPs (indicated with asterisks). (**a**) Markers SNP1382 (iniaGBS52443) and SNP1383 (iniaGBS52444), located in an intergenic region, and which were significant for both WSCCa and WSCCa/m^2^. (**b**) Marker SNP1507 (iniaGBS56801), part of a gene encoding a serine/threonine-protein kinase, which was significant for WSCa.

**Figure 4 plants-10-00539-f004:**
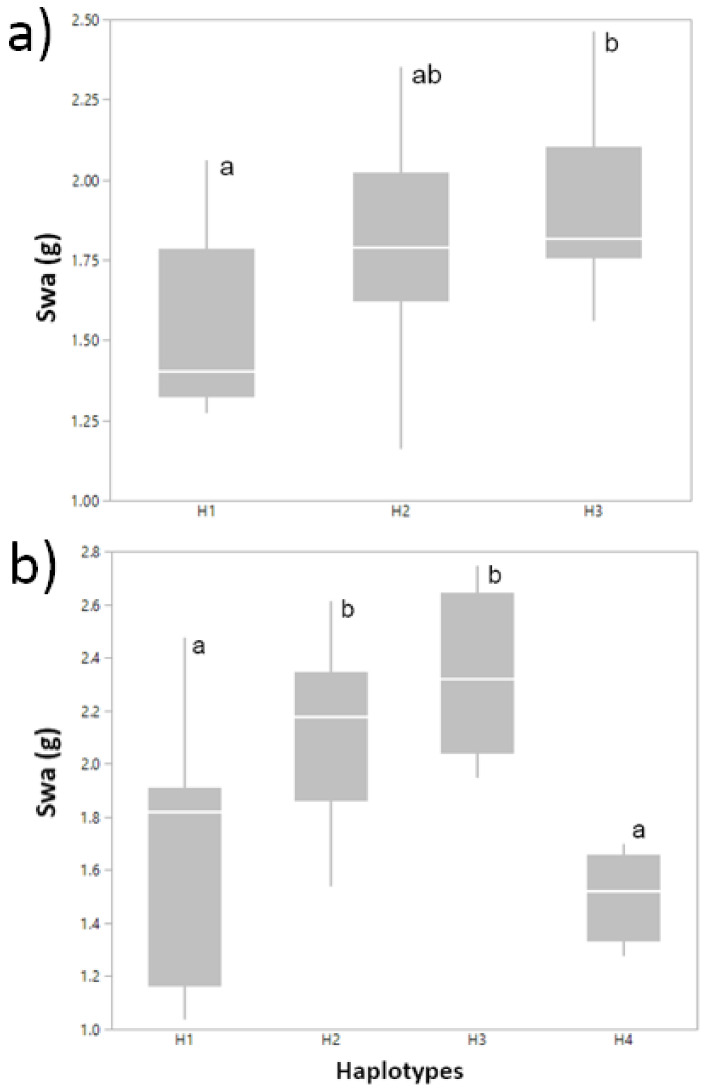
Representative effects of significant haplotypes on SWa. (**a**) Haplotypes H1–H3 identified for Santa Rosa, 2011. (**b**) Haplotypes H1–H4 identified for Cauquenes 2012. Details for haplotypes are included in Table 4. Different letters indicate significant differences between haplotype LSmeans (*p*-value < 0.05, Tukey Honestly Significant Difference [HSD] test).

**Table 1 plants-10-00539-t001:** ANOVA and estimate of variance components for traits evaluated in 225 genotypes at Cauquenes (water-limited) and Santa Rosa (full irrigation) sites, in 2012.

Trait		ANOVA (*p*-Value)			Variance Components (% *σ*^2^*_P_*)			Ratio	
	Genotype	Site	Rep(Site)	Genotype X Site	*σ* ^2^ *_G_*	*σ* ^2^ *_GxS_*	*σ* ^2^ *_e_*		*σ* ^2^ *_G_/σ* ^2^ *_P_*
GY	<0.0001	<0.0001	<0.0001	0.0002	16.1	15.3	68.6		0.16
SWa	<0.0001	<0.0001	0.147	0.2303	51.6	1.9	46.5		0.52
SWm	<0.0001	0.88	0.2302	<0.0001	46.6	12.4	41.0		0.47
WSCa	0.0115	<0.0001	0.1025	0.3557	5.8	1.9	92.3		0.06
WSCm	0.0003	<0.0001	0.0032	0.0279	5.0	9.4	85.7		0.05
WSCCa	<0.0001	<0.0001	0.0527	0.2968	29.0	2.0	69.1		0.29
WSCCm	<0.0001	<0.0001	0.012	0.0003	17.9	15.3	66.8		0.18
WSCCa/m^2^	<0.0001	<0.0001	0.0006	0.0001	5.8	18.3	75.9		0.06
WSCCm/m^2^	<0.0001	<0.0001	0.0943	0.3662	13.5	1.5	84.9		0.14
ARWSC	<0.0001	<0.0001	0.0874	0.9364	19.5	0.0	80.5		0.20

GY, grain yield; SWa, stem weight at anthesis; SWm, stem weight at maturity; WSCa, water-soluble carbohydrate concentration at anthesis (mg/g); WSCm, water-soluble carbohydrate concentration at maturity (mg/g); WSCCa, total WSC content at anthesis (mg/stem); WSCCm, total WSC content at maturity (mg/stem); WSCCa/m^2^, WSCCa per area unit; WSCCm/m^2^, WSCCm per area unit; ARWSCs, apparent remobilization of water-soluble carbohydrates; GxS, genotype by site interaction.

**Table 2 plants-10-00539-t002:** Number of single-SNP markers significantly associated (*p*-value < 0.001) with grain yield and stem WSC traits, in the different assessed conditions.

Trait ^a^			Site/Condition ^b^			
	Cau 2011	Cau 2012	SRosFI2012	SRosFI2011	SRosMWS2011	Total
GY					4	4
SWa					2	2
SWm		1			1	2
WSCa		22	1			23
WSCm		2	3			5
WSCCa		1	1	5		7
WSCCm	3		5		3	11
WSCCa_m2			1	7	7	15
WSCCm_m2	2		1	6		9
ARWSC				18		18
Total	5	26	12	36	17	96

^a^ Traits: GY, grain yield; SWa, stem weight at anthesis; SWm, stem weight at maturity; WSCa, water-soluble carbohydrate concentration at anthesis (mg/g); WSCm, water-soluble carbohydrate concentration at maturity (mg/g); WSCCa, total WSC content at anthesis (mg/stem); WSCCm, total WSC content at maturity (mg/stem); WSCCa/m^2^, WSCCa per area unit; WSCCm/m^2^, WSCCm per area unit; ARWSC, apparent remobilization of water-soluble carbohydrates. ^b^ Site/condition: Cau, Cauquenes (water-limited site); SRosFI2011 and SRosFI2012, seasons 2011 and 2012, Santa Rosa full irrigation; SRosMWS2011, Santa Rosa mild water stress, season 2011.

**Table 3 plants-10-00539-t003:** List of selected SNP-markers associated with the assessed traits. A complete list is included in Appendix A
Appendix A.

Trait ^a^	Site/Condition ^b^	Marker	SNP	Chr.	*p*-Value	*q*-Value	*R* ^2^	Position	Protein	Functional Category	Ensembl Gene ID
WSCa	Cau 2012	iniaGBS56801	C/T	3A	2.5 × 10^−5^	0.0818	0.1232	Gene	serine/threonine-protein kinase	Protein synthesis/modification	TRIAE_CS42_3AS_TGACv1_212383_AA0700440
	Cau 2012	iniaGBS12206	C/T	4B	0.0002	0.1136	0.1014	Gene	Uncharacterized		TRIAE_CS42_4BL_TGACv1_320632_AA1045170
	Cau 2012	iniaGBS21514	A/G	5D	0.0003	0.1136	0.0981	Gene	Guanosine nucleotide diphosphate dissociation inhibitor	Energy/metabolism	TRIAE_CS42_5DS_TGACv1_456722_AA1476920
	Cau 2012	iniaGBS13140	A/T	5B	0.0004	0.1247	0.0908	Gene	Strictosidine synthase	Energy/metabolism	TRIAE_CS42_5BL_TGACv1_404735_AA1309530
	Cau 2012	iniaGBS61502	A/G	1A	0.0007	0.1247	0.0837	Gene	ABC transporter C family member 10	Transport	TRIAE_CS42_1AL_TGACv1_000028_AA0000820
	Cau 2012	iniaGBS86062	C/G	6B	0.0007	0.1247	0.0861	Gene	Hypersensitive induced response protein 3	Cell disease/defense	TRIAE_CS42_5BL_TGACv1_404707_AA1308960
ARWSC	SRosFI2011	iniaGBS83719	G/T	6D	2.3 × 10^−6^	0.0017	0.1491	Intergenic	Non-coding		
	SRosFI2011	iniaGBS76557	G/T	6D	6.8 × 10^−6^	0.0034	0.1373	Gene	Uncharacterized		TRIAE_CS42_U_TGACv1_644569_AA2140450
WSCCa/m2	SRosFI2011	iniaGBS52443	A/G	6B	0.0001	0.1000	0.1248	Intergenic	Non-coding		
	SRosFI2011	iniaGBS52444	G/T	6B	6.4 × 10^−5^	0.1000	0.1248	Intergenic	Non-coding		
	SRosMWS201	iniaGBS24966	A/C	7B	0.0002	0.3181	0.0993	Gene	Endoglucanase	Energy/metabolism	TRIAE_CS42_7BS_TGACv1_592132_AA1931340

^a^ Traits: WSCa, water-soluble carbohydrate concentration at anthesis; ARWSC, apparent remobilization of water-soluble carbohydrates; WSCCa/m^2^, total WSC content at anthesis per area unit. ^b^ Site/condition: Cau, Cauquenes (water-limited site); SRosFI, Santa Rosa full irrigation; SRosMWS, Santa Rosa mild water stress.

**Table 4 plants-10-00539-t004:** Haplotypes identified for selected genome regions containing significant single SNP-markers.

Site/Condition ^a^	Trait	Position/Gene	*GSS* ^b^	*p*-Value	*q*-Value	Significant Haplotypes	Freq.	Code ^c^
SRosMWS2011	WSCa	endoglucanase gene	20.4	0.0024	0.0210	AGACCCC	0.09	
	WSCCa/m^2^	endoglucanase gene	18.9	0.0043	0.0220	CACGTTG	0.03	
Cau2012	SWa	serine/threonine	30.3	0.0043	0.0205	GTGTTCAT	0.07	H1
						GTAGCTAC	0.07	H2
						TTAGCCAT	0.03	H3
						GTAGCTGT	0.03	H4
	SWm	serine/threonine	23.2	0.0395	0.0403	GCGTCCAT	0.03	
	WSCCm	serine/threonine	13.0	0.0403	0.0403	GCGTCCAT	0.03	
	WSCCa/m^2^	serine/threonine	16.5	0.0102	0.0205	GTAGCTGT	0.03	
SRosFI2011	SWa	intergenic region	20.4	0.0024	0.0215	ACACCAGGGCCT	0.04	H1
						ACACCAGAACCT	0.36	H2
						CCGTTAGGGTTC	0.11	H3
	SWm	intergenic region	18.9	0.0043	0.0210	ACACCAGGGCCT	0.04	
						ACACCAGAATCT	0.28	
						CCGTTAGGGTTC	0.11	
	WSCCm/m^2^	intergenic region	12.9	0.0444	0.0494	ACACCAGGGCCT	0.04	

^a^ Site/condition: SRosFI, Santa Rosa full irrigation; SRosMWS, Santa Rosa mild water stress. ^b^ GSS, global score statistic. ^c^ Haplotype number for haplotypes identified for Cauquenes, 2012, and Santa Rosa, 2011 (depicted in Figure 4).

## Data Availability

The data are available from the corresponding author upon reasonable request.

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
