# Peer review of "Genome-Wide Association of Stem Carbohydrate Accumulation and Remobilization during Grain Growth in Bread Wheat (Triticum aestivum L.) in Mediterranean Environments"

_plants, 2021, doi:10.3390/plants10030539_

Round 1

Reviewer 1 Report

remobilization during grain growth in bread wheat (Triticum aestivum L.) in Mediterranean environments” by Guerra et al. is an interesting study and authors tried to explain the story with some justification. However, there are few comments which should be tackled before recommendation. I would suggest following changes:

Abstract

Page 1 line 21-23, the sentence is not clear. Please rewrite this sentence to clarify the meanings.

Page 1 line 25-56, the sentence is incomplete.

Introduction

Page 1 line 42. Please include the following references:

  1. Shokat, S; Großkinsky, DK; Liu, F; 2021. Impact of elevated CO2 on two contrasting wheat genotypes exposed to intermediate drought stress at anthesis. Journal of Agronomy and Crop Science. 207: 20-33.
  2. Shokat, S; Großkinsky, DK; Roitsch, T; Liu, F; 2020. Activities of leaf and spike carbohydrate-metabolic and antioxidant enzymes are linked with yield performance in three spring wheat genotypes grown under well-watered and drought conditions. BMC Plant Biology. 20.400

Page 2, line 48-49, please elaborate this sentence. 

Page 2, line 58, please add a reference after 23:

Shokat, S; Sehgal, D; Vikram, P; Liu, F; Singh, S; 2020. Molecular markers associated with agro-physiological traits under terminal drought conditions in bread wheat. International Journal of Molecular Sciences. 21(9): 3156.

Page 2, lines 72-77, this sentence need clarity. It is better to divide it into two sentences.

Page2, lines 83-84, please delete the highlighted sentence “in the same germplasm collection”

Page2, lines 93, It is better to represent yield in kg ha-1

Results

Page 5

Lines 140-150 needs clarity. Please rewrite to explain the meanings.

Please remove “as” from line 145.

Page 7, lines 11-13 need to rewrite.

Page 8, line 29, is it important to write “Endoglucanase”?

Page 11, line 137, is it important to write first alphabet in capital letters?

What was length of markers in terms of base pairs. Is it fair to use SNPs for blast analysis or gene detection?

Materials and Methods

How terminal drought stress was achieved? Please indicate in growth conditions

Author Response

Remobilization during grain growth in bread wheat (Triticum aestivum L.) in Mediterranean environments” by Guerra et al. is an interesting study and authors tried to explain the story with some justification. However, there are few comments which should be tackled before recommendation. I would suggest following changes:

-Reply (R): Thank you very much for your review. We appreciate your suggestions, which contributed to improving the first version of our paper. We modified the text according to your indications. Changes are highlighted in blue in the Word file containing the main manuscript.

Abstract

Page 1 line 21-23, the sentence is not clear. Please rewrite this sentence to clarify the meanings.

-R: The sentence was rewritten.

Page 1 line 25-56, the sentence is incomplete.

-R: The sentence was modified.

Introduction

Page 1 line 42. Please include the following references:

  1. Shokat, S; Großkinsky, DK; Liu, F; 2021. Impact of elevated CO2 on two contrasting wheat genotypes exposed to intermediate drought stress at anthesis. Journal of Agronomy and Crop Science. 207: 20-33.
  2. Shokat, S; Großkinsky, DK; Roitsch, T; Liu, F; 2020. Activities of leaf and spike carbohydrate-metabolic and antioxidant enzymes are linked with yield performance in three spring wheat genotypes grown under well-watered and drought conditions. BMC Plant Biology. 20.400

-R: Both references were included in the revised version.

Page 2, line 48-49, please elaborate this sentence. 

-R: The sentence was rewritten.

Page 2, line 58, please add a reference after 23:

Shokat, S; Sehgal, D; Vikram, P; Liu, F; Singh, S; 2020. Molecular markers associated with agro-physiological traits under terminal drought conditions in bread wheat. International Journal of Molecular Sciences. 21(9): 3156.

-R: The reference was included in the revised version.

Page 2, lines 72-77, this sentence need clarity. It is better to divide it into two sentences.

-R: The sentence was rewritten.

Page2, lines 83-84, please delete the highlighted sentence “in the same germplasm collection”

-R: The sentence was modified accordingly.

Page2, lines 93, It is better to represent yield in kg ha-1

-R: Yield was expressed in kg ha-1. The corresponding text and Fig.1 were modified incorporating this unit.

Results

Page 5

Lines 140-150 needs clarity. Please rewrite to explain the meanings.

-R: The text was rewritten.

Please remove “as” from line 145.

-R: This change was incorporated.

Page 7, lines 11-13 need to rewrite.

-R: The sentence was rewritten.

Page 8, line 29, is it important to write “Endoglucanase”?

-R: No. Capital letter was changed.

Page 11, line 137, is it important to write first alphabet in capital letters?

-R: No. Capital letters were changed.

What was length of markers in terms of base pairs. Is it fair to use SNPs for blast analysis or gene detection?

-R: Each SNP was in the middle of a 64 bp region. This length is usually considered adequate to perform BLAST (even shorter sequences are utilized to analyze PCR primers). Gene detection was done in two steps. First, we utilized our previously mapped position for each SNP region, and then searched it in the wheat genome sequence (available at the Ensemble platform). Second, we ran a BLAST for confirming the match, taking the lowest e-value.

Materials and Methods

How terminal drought stress was achieved? Please indicate in growth conditions

-R: This information was included in the revised manuscript (Materials and Methods section).

Reviewer 2 Report

The paper is well written, clear English style, easy to read and to understand. The studied material was used before for many other purposes which were published in other high ranking journals. The topic accumulation and remobilization of water-soluble carbohydrates (WSC) in stems is very relevant for increasing yield and reducing negative effects of water deficit. Many other papers about that topic are available which are cited. New are the environmental conditions of Mediterranean climate in central Chile. This other geographical area shows interesting results to broaden the knowledge about the marker-trait associations in that topic and to identify candidate genes. The conclusions are based on the results and consistent. The topic of the paper is well described.

Author Response

The paper is well written, clear English style, easy to read and to understand. The studied material was used before for many other purposes which were published in other high ranking journals. The topic accumulation and remobilization of water-soluble carbohydrates (WSC) in stems is very relevant for increasing yield and reducing negative effects of water deficit. Many other papers about that topic are available which are cited. New are the environmental conditions of Mediterranean climate in central Chile. This other geographical area shows interesting results to broaden the knowledge about the marker-trait associations in that topic and to identify candidate genes. The conclusions are based on the results and consistent. The topic of the paper is well described.

- Reply: Thank you very much. We appreciate your comments.